# The Epidemiology of Shoulder Injuries in Water Polo Players: A Monocentric Descriptive Study on Clinical and Radiological Presentation

**DOI:** 10.3390/jcm13071951

**Published:** 2024-03-27

**Authors:** Marco Minelli, Umile Giuseppe Longo, Riccardo Ranieri, Federico Pascucci, Filippo Giunti, Marco Conti, Francesco Catellani, Alessandro Castagna

**Affiliations:** 1Department of Biomedical Sciences, Humanitas University, Via Rita Levi Montalcini 4, Pieve Emanuele, 20090 Milan, Italy; marcomariaminelli@gmail.com (M.M.); dr.riccardoranieri@gmail.com (R.R.); federico.pascucci@st.hunimed.eu (F.P.); filippo.giunti@st.hunimed.eu (F.G.); francesco.catellani@gavazzeni.it (F.C.); acastagna@me.com (A.C.); 2Fondazione Policlinico Universitario Campus Bio-Medico, Via Alvaro del Portillo 200, 00128 Roma, Italy; 3Research Unit of Orthopaedic and Trauma Surgery, Department of Medicine and Surgery, Università Campus Bio-Medico di Roma, Via Alvaro del Portillo, 21, 00128 Roma, Italy; 4Shoulder and Elbow Unit, Department of Orthopedic and Trauma Surgery, Humanitas Clinical and Research Center, IRCCS, Via Manzoni 56, 20089 Milan, Italy; conti.marco@gmail.com; 5Humanitas Gavazzeni, 24125 Bergamo, Italy

**Keywords:** shoulder, water polo, players, injury, magnetic resonance imaging

## Abstract

Water polo players’ shoulders are exposed to repeated overhead and throwing motions as well as direct and indirect traumas. Shoulder injuries account for over half of all injuries sustained by water polo players. This is a monocentric descriptive epidemiological study on the clinical and radiological presentation of a consecutive series of water polo players from January 2002 to September 2022. All patients underwent clinical and physical examinations and an MRI arthrogram. A total of 92 water polo players were included in this study. Fifty-three patients (57.6%) reported at least one previous shoulder instability episode; 100% of patients in this group were diagnosed with a capsulolabral complex lesion, and 88.7% of these players complained of subjective symptoms of shoulder instability (RR: 4.4). A total of 39 out of 92 patients (42.4%) did not report previous shoulder dislocation episodes; the mean age at presentation in this group was significantly higher than the mean age of the patients who experienced previous instability episodes (*p* < 0.01), and the throwing arm was affected in 79.5% of patients (RR = 1.41).

## 1. Introduction

Water polo was the first team sport to be introduced at the modern Olympic Games in 1900 [1]. Traditionally popular in European countries, this sport’s popularity is growing in several areas of the world [2,3]. Water polo is a water-based contact sport that involves swimming, throwing, and defending [4]. Water polo is a physically demanding sport, involving intense sprint swimming bursts to change direction every 6.2 s [5].

In particular, shoulders are exposed to intense forces. Passing and shooting are performed from end-of-range shoulder abduction and external rotation at arm speeds of up to 24.1 ± 1.58 ms^−^^1^ (86.8 ± 5.69 kmh^−^^1^) [6]. Resultant joint torques appear to be similar to the ones observed in other overhead sports, such as baseball and American football, producing an internal impingement of rotator cuff tendons, superior capsulolabral complex, and a long head of the biceps (LHB) anchor between the greater tuberosity and postero-superior glenoid articular surfaces [7,8,9,10]. In addition, repeated overhead and throwing motions appear to result in adaptations in the throwing arm’s range of motion, such as increased external rotation and decreased internal rotation [7,8,9,10,11]. Furthermore, water polo players use an adapted upright swimming posture to allow for the movement of the ball and a clear view of the opposition, resulting in shoulder abduction and consequent stress upon the rotator cuff tendons [4,7,12]. Water polo players keep their arms above their heads in order to physically obstruct opposing players, exposing the shoulder joint to external forces [13,14]. Moreover, with water polo being a contact sport, players are exposed to direct and indirect traumatic impacts, potentially resulting in acute shoulder events, such as shoulder dislocations and/or subluxations [4,13,14,15,16].

Thus, water polo players are vulnerable to both acute traumatic events and overuse injuries from repetitive swimming and overhead throwing; indeed, shoulder, head/neck, wrist/hand/fingers, elbow, lumbar, knee, and hip injuries are common [17]. However, the most frequently injured site during the 2013 FINA World Championships was the shoulder [14]. Shoulder injuries account for up to 51% of all injuries in water polo players, and period prevalence values of up to 98.4% have been reported in cross-sectional studies [4,13,17,18]. The aim of this study was to describe clinical and radiological presentation of shoulder injuries in water polo players.

## 2. Materials and Methods

This was a monocentric descriptive epidemiological study on the clinical and radiological presentation of a consecutive series of water polo players over a 20-year period. A single experienced shoulder surgeon performed all of the clinical and radiological diagnoses. This study adhered to the principles outlined in the Declaration of Helsinki and was approved by the IRCCS Independent Ethics Committee.

Inclusion criteria encompassed Italian amateur and professional water polo players competing in national and international leagues who had a consultation at our institution from January 2002 to September 2022. Patients were included in this study in the case of shoulder pain and/or functional limitations impeding engagement in water polo-related activities, recurring traumatic glenohumeral instability, and first-time instability episodes (gleno-humeral dislocation or subluxation). No exclusion criteria were applied, except in cases where individuals were unwilling or unable to participate in the study.

During clinical examination, patients’ symptoms and medical histories were investigated. Patients were asked about shoulder pain and/or subjective perceptions of shoulder instability (apprehension and/or weakness under load); specifically, they were asked when these symptoms started, whether these occurred in the throwing or in the non-throwing arm, and whether these occurred during activity or at rest. Then, previous dislocation and/or subluxation (and related direction/s) episodes were investigated.

Upon physical bilateral and comparative examination, active (AROM) and passive (PROM) range of movements were evaluated. Particularly, forward flexion, abduction, and internal and external rotation with the arm to the side of the body were investigated. An internal rotation difference greater than 20 degrees between the two shoulders was defined as gleno-humeral rotation deficit (GIRD) after exclusion of concomitant pathologies [19]. Whenever correctly evaluable, provocative and rotator cuff tests were performed. In particular, provocative tests reproduce the dislocation mechanism, eliciting apprehension, a painful and uncomfortable reaction in the patient. In the case of anterior instability, the typical maneuver consists of combined solicitation via abduction/horizontal extension/external rotation of the upper limb (apprehension test) [20], while in the case of posterior instability, axial solicitation of the horizontal flexed upper limb reproduces the dislocation mechanism [21]. In order to evaluate rotator cuff function, specific tests were carried out. In particular, supraspinatus function was evaluated by opposing patient active abduction (Jobe test); infraspinatus and teres minor function was evaluated by opposing active external rotation (Patte test), and subscapular function was evaluated by opposing active internal rotation (lift-off and belly-press test) [22]. Scapular dyskinesis was classified according to Kibler et al. [23].

All patients underwent an MRI arthrogram in a routine multiplanar sequence (axial, coronal-oblique, and sagittal) to identify potential injuries. Labral lesions were classified according to location into anterior, combined anterior and superior, isolated superior and LHB anchor, isolated posterior, and combined anterior and posterior. Cuff lesions were classified according to the amount of retraction (Patte classification [24]) and fatty infiltration (Goutallier classification [25]). Gleno-humeral and/or acromioclavicular chondral damage and/or osteoarthritic bone degeneration were reported whenever present.

Final diagnosis was based upon history evaluation, physical examination, and MRI arthrogram findings.

## 3. Results

A total of 92 water polo players (80 males, 87%; 12 females, 13%) were included in this study. Thirty-eight were professionals and fifty-four were amateur water polo players. The mean age was 27.7 years (range 16–60 years).

### 3.1. Clinical Presentation

A total of 68 (74%) patients complained of shoulder pain; in 53 cases (58%), the symptoms were related to water polo activity, while in 15 (16%) patients, the symptoms also occurred at rest. A total of 49 (53%) patients reported symptoms of shoulder instability; in 34 players (37%), these symptoms were related to water polo activity, while in 15 (16%) cases, they also occurred at rest. Fifty-three patients (57.6% of the total) reported at least one previous gleno-humeral instability episode (52 anterior dislocations and/or subluxations, 1 posterior).

Upon a physical examination, in 27 patients, internal rotation was observed to be decreased compared to the contralateral side. In 1 case, this was due to a post-traumatic adhesive capsulitis, while the other 26 cases (28%) were classified as GIRD. The Jobe test was positive in fourteen patients (15% of the total). The infraspinatus, teres minor, and supscapularis-specific tests were negative in all patients. Sixty-four patients (70%) had a positive apprehension test. One case of Kibler grade 2 scapulo-toracic dyskinesia was observed.

### 3.2. Radiological Presentation

An MRI arthrogram revealed capsulolabral complex lesions in 78 patients (84.8%). In particular, the MRI arthrogram detected 28 isolated anterior labrum lesions (30.4% of the total; in 2 cases, a combined fracture of the inferior glenoid was observed), 17 isolated superior labrum and LHB anchor lesions (18.5%), 12 combined anterior and superior labrum lesions (13.0%), 6 combined anterior labrum lesions and supraspinatus tears (6.5%), 5 anterior labroligamentous periosteal sleeve avulsions (ALPSA, 5.4%), 2 humeral avulsions of glenohumeral ligaments (HAGL, 2.2%), 6 combined anterior and posterior labrum lesions (6.5%) and 1 combined superior labrum and supraspinatus lesion (1.1%). Only one isolated posterior capsulolabral complex lesion was detected. Seven isolated supraspinatus tears (7.6%) and 2 supraspinatus calcific enthesopathies without any evidence of tears (2.2%) were detected; fatty infiltration was classified as Goutallier grade 1 in seven out of seven supraspinatus tears, while tendon retraction was classified as Patte grade 1 in five patients and as Patte grade 2 in two patients. Four gleno-humeral chondropathy cases (4.34%) were observed.

The patients were divided into groups according to the MRI arthrogram findings. The mean age, laterality, previous instability episodes, symptomatology, and physical examination findings for each group are shown in Table 1.

Fifty-three out of ninety-two patients (57.6%) reported at least one previous shoulder dislocation and/or subluxation episode. The mean age in this group was 24.5 years. The throwing arm was affected in 56.6% of cases. A total of 29 patients (54.7%) complained of shoulder pain, and 47 cases (88.7%) reported symptoms of shoulder instability. The apprehension test was positive in 50 patients (94%). In this group, the MRI revealed 26 isolated anterior labrum lesions (49.0%), 9 combined anterior and superior labrum lesions (17%), 5 combined anterior and posterior labrum lesions (9.4%), 5 ALPSAs (9.4%), 4 combined anterior labrum and supraspinatus lesions (7.6%), 2 HAGLs (3.8%), and 2 isolated superior labral and LHB anchor lesions (3.8%).

Thirty-nine out of ninety-two patients (42.4%) did not report previous shoulder dislocation episodes. The mean age in this group was 32 years. The throwing arm was affected in 79.5% of cases. In this group, 100% of patients complained of shoulder pain. Eight patients (21%) reported symptoms of shoulder instability. In this group, the MRI revealed 15 isolated superior labrum and LHB lesions (38.5%), 7 isolated supraspinatus lesions (17.9%), 4 cases of gleno-humeral chondropathy (10.3%), 3 combined anterior and superior labrum lesions (7.6%), 2 supraspinatus calcific tendinopathies (5.1%), 2 combined anterior and posterior labrum lesions (5.1%), 2 anterior labrum lesions (5.1%), 2 combined anterior labrum and supraspinatus lesions (5.2%), and 1 combined superior labrum and supraspinatus lesion (2.6%).

## 4. Discussion

Based on the main presenting symptoms, the physical examination, and the MRI findings, water polo players’ shoulders can be classified into two categories: painful and unstable shoulders.

Among the 39 patients who did not report previous instability episodes (42.4%), 46.1% were diagnosed with superior labrum and LHB anchor involvement, 23.1% with rotator cuff pathologies (supraspinatus calcific enthesopathies or tears), and 2.6% with combined superior glenoid labrum and supraspinatus lesions (total 71.8%). As previously described by Feltner et al. and Melchiorri et al. [6,8], water polo players pass and shoot through overhead and throwing motions; the resulting joint torques produce an internal impingement of the rotator cuff tendons, superior glenoid capsulolabral complex, and LHB anchor [7,8,9,10]. Indeed, internal impingement results in structural injuries, bringing about chronic pain and discomfort in the shoulder region; in this group, 100% of patients complained of shoulder pain (RR = 1.85) [10]. The mean age (32 years) at presentation in this group was significantly higher than the mean age (24.5 years) of the patients who experienced acute dislocations (*p* < 0.01), and the throwing arm was affected in 79.5% of patients with no previous dislocation (RR = 1.41). This suggests that water polo players who had no previous shoulder dislocations developed shoulder lesions over time as a consequence of repetitive overhead motions, resulting in chronic overload [7,8,9,10,11]. In the patients who had no previous dislocations, superior labrum and LHB anchor involvement was the most frequent finding. Interestingly, 82.4% of isolated superior capsulolabral complex and LHB anchor lesions were detected in players’ throwing arms, and 94.1% of patients suffering from this lesion complained of chronic shoulder pain. Moreover, 47.1% of patients in this group presented a difference greater than 20 degrees in internal rotation between the two shoulders. Indeed, Keller et al. and Kibler et al. [11,19] described how repeated overhead and throwing motions result in modifications in the throwing arm’s range of motion, particularly decreased passive internal rotation, defined as gleno-humeral internal rotation deficit (GIRD). 

Fifty-three patients (57.6% of the total) experienced a previous instability episode. As previously described by Franić et al. and Hams et al. [2,4], water polo players defend their teams by keeping their arms above their heads in order to physically obstruct opposing players, exposing maximally abducted and externally rotated shoulders to external forces [13,14,15,16]. Furthermore, players are exposed to direct and indirect traumas, potentially resulting in shoulder instability episodes, such as shoulder dislocations and/or subluxations [4,13,14,15,16]. The mean age (24.5 years) at presentation was significantly lower than the mean age (32.5 years) of the patients who suffered from chronic overload injuries (*p* < 0.01). Shoulder dislocation results in damage to the capsulolabral complex, which plays a fundamental role in maintaining shoulder stability. Fifty-three out of fifty-three (100%) patients in this group were diagnosed with a capsulolabral complex lesion, and 88.7% of these players complained of subjective symptoms of shoulder instability (RR: 4.4).

This study presents several limitations. First, being a retrospective descriptive study, it cannot be used to establish cause-and-effect relationships between variables. Second, focusing only on clinical and radiological presentation, it does not cover therapeutic choices and outcomes. In particular, therapeutic outcomes, such as return to sport, are what both professional and amateur athletes care the most about. Additionally, the patients were not stratified into different groups according to whether they were amateurs or professionals; these categories have different workloads, which may play an etiopathogenetic role in water polo players’ shoulder injuries [7,12]. Additionally, a mechanism of injury analysis was not carried out. Moreover, physical examination has non-negligible intra- and inter-observer variability, so it is not enough to reach a definitive diagnosis. Furthermore, for most of the described lesions, a definitive diagnosis can only be reached via direct visualization and dynamic tests during arthroscopic surgery, which were not described in this study. Eventually, no functional score was calculated; even if their main function is to compare pre- and post-interventional status in experimental studies, functional scores could have helped in better describing the clinical presentation.

The shoulder is reported to be the most commonly injured site in water polo players [4,13,14,15,16,17,18]. Even though this is only a retrospective descriptive study, to the best of our knowledge, this work is currently the largest epidemiological study on the clinical and radiological presentations of shoulder injuries in these athletes.

## Figures and Tables

**Table 1 jcm-13-01951-t001:** Clinical presentations of water polo players according to different MRI arthrogram findings.

**MRI Arthrogram Findings**	**#**	**Mean Age**	**Throwing Arm**	**Instability Episodes**	**Apprehension**	**GIRD**	**Pain**	**Instability**
Isolated anterior labrum	28	23.5 years old	67.8%	92.9%	96.3%	11.1%	35.7%	89.3%
Combined anterior and superior labrum	12	23.2 years old	25.0%	75.0%	83.3%	25.0%	83.3%	66.7%
Combined anterior labrum + supraspinatus	6	43.8 years old	83.4%	66.7%	100%	0.00%	100%	83.4%
ALPSA	5	27.8 years old	60%	100%	100%	20.0%	60.0%	100%
HAGL	2	29.5 years old	100%	100%	100%	50%	50.0%	100%
Isolated superior labrum and LHB anchor	17	26.5 years old	82.4%	11.8%	52.9%	47.1%	94.1%	29.4%
Combined superior labrum + supraspinatus	1	45.2 years old	100%	0.00%	100%	0.00%	100%	0.00%
Combined anterior + posterior labrum	6	27.7 years old	66.7%	83.4%	100%	16.7%	66.7%	83.4%
Isolated posterior labrum	1	19.3 years old	100%	0.00%	0.00%	0.00%	100%	100%
Isolated supraspinatus	7	42.2 years old	85.7%	0.00%	0.00%	14.3%	100%	0.00%
Calcific tendinopathy	2	40.1 years old	100%	0.00%	0.00%	100%	100%	0.00%
Chondropathy	4	54.5 years old	100%	0.00%	/	/	100%	0.00%

## Data Availability

The datasets used and/or analyzed during the current study are available from the corresponding author upon reasonable request.

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
