# Peer review of "The Epidemiology of Shoulder Injuries in Water Polo Players: A Monocentric Descriptive Study on Clinical and Radiological Presentation"

_jcm, 2024, doi:10.3390/jcm13071951_

Round 1
Reviewer 1 Report
Comments and Suggestions for Authors
introduction: talk more about common injuries in polo water and percentages if present
Materials and methods : how many evaluators was involved , it can provoke interobserver differences
Lind 93 : follow up range ? or STD
Line 98 : what is the range of age?
Lind 109 : references not indicated in results
percentages should be precised if it was more common in professional or amateur ? need to do subgroup analysis ... return to sport , physiotherapy protocol is completely different between amateurs and professional as well as level of participation and need .. how this could be evaluated ?
Line 184: 5 professional and 8 amateurs decided for conservative treatment : how this was chosen ? is it the end of their career?
professional player went more for arthroscopic surgery , it's sure that insurance / or wish to return to competition ... is there a bias of chosen treatment ?
Subgroup analysis is a must
results are too long , should be summerized in tables
Discussion: need to stress more on literatures
Reviewer 2 Report
Comments and Suggestions for Authors
This is a very interesting field, but I felt the results part is very confusing.
It's just a list of case reports.
I think it should be grouped by several broad categories. For example, they are shoulder dislocation, labrum tear and rotator cuff tear, etc.
And table 1 is also difficult to understand.
Can you add a photo of the actual playing field that shows the shoulder position clearly?
Please compare other sports such as like handball, swimming, and etc.
Furthermore, if you could also show the disorders in other parts of the body in water polo, the reader would understand that shoulder disorders are common in water polo.
Reviewer 3 Report
Comments and Suggestions for Authors
Very interesting study!
Here are some questions to be cleared:
-Please rewrite the abstract, at the moment it is only describing the methods and the results.
-Introduction: well written
-Methods: Please describe which rotator cuff tests were used. Were MRIs performed with or without a contrast medium?
-results: which were antero-inferior and which posterior dislocations? Please restructure and rewrite the results section. One cannot follow the long list of unstructured results. I would suggest to use subheadings.
-which measurements did you apply on the glenoid/the humeral head/joint to assess radiological instability factors?(such as Glenohumeral Joint ant/ post Translation, Hill-Sachs Lesion Measurement, Glenoid Bone Loss, Instability Index, Glenoid Version, Humeral Retroversion, Critical Shoulder Angle (CSA), Bony Bankart Lesion Measurement).
-since there is such a big upcoming of re-dislocations, the study group should analyze the differences between patients who had persistent instability and those who didn't.
-good conclusion!
-please optimize the graphics
Comments on the Quality of English Language
Please improve Quality of English Language.
Round 2
Reviewer 2 Report
Comments and Suggestions for Authors
I think it is well rewritten and addresses my comments.
I think this paper is acceptable.
Author Response
thank you!